# Identification of microsporidia host-exposed proteins reveals a repertoire of rapidly evolving proteins

Aaron W. Reinke[1], Keir M. Balla[1], Eric J. Bennett[1] & Emily R. Troemel[1]

Pathogens use a variety of secreted and surface proteins to interact with and manipulate their hosts, but a systematic approach for identifying such proteins has been lacking. To identify these 'host-exposed' proteins, we used spatially restricted enzymatic tagging followed by mass spectrometry analysis of *Caenorhabditis elegans* infected with two species of *Nematocida* microsporidia. We identified 82 microsporidia proteins inside of intestinal cells, including several pathogen proteins in the nucleus. These microsporidia proteins are enriched in targeting signals, are rapidly evolving and belong to large *Nematocida*-specific gene families. We also find that large, species-specific families are common throughout microsporidia species. Our data suggest that the use of a large number of rapidly evolving species-specific proteins represents a common strategy for microsporidia to interact with their hosts. The unbiased method described here for identifying potential pathogen effectors represents a powerful approach to study a broad range of pathogens.

[1] Division of Biological Sciences, Section of Cell and Developmental Biology, University of California, San Diego, 9500 Gilman Drive, La Jolla, California 92093, USA. Correspondence and requests for materials should be addressed to A.W.R. (email: awreinke@gmail.com).

Pathogens exploit hosts to promote their own proliferation. Viral, bacterial and eukaryotic pathogens control their hosts using effector proteins that interact directly with host molecules[1–3]. These effector proteins can be exported out of the pathogen into host cells or they can remain attached to the pathogen but with regions of the protein exposed to the host environment. These host-exposed proteins perform molecular functions that range from manipulation of host defenses to modulation of host pathways that can promote pathogen growth[4,5]. In many cases these proteins are evolving under diversifying selection, such that variation among these proteins can influence host survial[6–8]. Examples to date indicate considerable variation in the proteins that pathogens use to interface with their hosts. The conservation of these host-exposed proteins varies among different types of pathogens. Whereas most effectors of a strain of Pseudomonas syringae are present in other Pseudomonas strains and over 35% are conserved in other bacterial genera[9], fewer than 15% of predicted host-exposed proteins of Plasmodium falciparum are reported to be conserved among Plasmodium species[3].

Comprehensive identification of pathogen proteins that are host-exposed is challenging, because they need to be distinguished from proteins that are localized inside of pathogen cells. Several studies have addressed this problem by identifying proteins secreted from pathogens into culture media[10,11]. However, such studies potentially miss proteins that are only present in the native context. To circumvent this issue, a recent study chemically labelled proteins inside pathogenic bacteria and then identified those that were delivered inside of host cells[12]. Although powerful, this approach requires that a pathogen be both culturable and genetically tractable, and thus it is not generally applicable to many intracellular pathogens. In addition, these approaches do not provide information on the subcellular localization for pathogen proteins within host cells. To address these limitations, we adapted spatially restricted enzymatic tagging for the study of pathogen host-exposed proteins. Spatially restricted enzymatic tagging is a recently developed approach for labelling proteins in specific subcellular locations. This approach uses the enzyme ascorbate peroxidase (APX) to promote biotin labelling of neighboring proteins, which can be subsequently purified and identified with mass spectrometry[13]. Here, we take advantage of this localized proteomics approach to identify host-exposed proteins from microsporidia that are localized in the intestinal cells of an infected animal.

Microsporidia constitute a large phylum of fungal-related obligate intracellular eukaryotic pathogens. The phylum contains over 1,400 described species that infect diverse animals including nematodes, arthropods and vertebrates, although individual species often have a narrow host range[14,15]. Dependent on their hosts for survival and reproduction, they have reduced genomes that lack several key regulatory and metabolic pathways[16,17]. Altogether these properties make microsporidia an excellent model of pathogen evolution. Despite the fact that microsporidia are of both medical and agricultural importance, tools for genetic modification of microsporidia are lacking and almost nothing is known about the proteins that enable interactions with their hosts[18].

Two potential targeting signals are known that could expose microsporidia proteins to the host. These are N-terminal signal-sequences that direct proteins for secretion[19], and transmembrane domains that could be used to attach proteins to the pathogen plasma membrane with regions of the microsporidia protein in direct contact with host molecules[20]. A number of studies have used these two targeting signals to predict the set of proteins encoded by pathogen genomes that are likely to be host-exposed[21,22]. However, it is unclear how accurate these approaches are at identifying such proteins in microsporidia

and these prediction methods do not distinguish between proteins partially or wholly outside the microsporidia cell from those directed to internal membranes or compartments[13]. Although some host-exposed microsporidia proteins have been characterized[23–29], no comprehensive identification of such proteins has been carried that include the intracellular stage of the pathogen.

Several microsporidia of the genus Nematocida naturally infect C. elegans, a model organism that offers a number of advantages for the study of host–pathogen interactions[30,31]. Infection of C. elegans by N. parisii begins with spores being ingested and then invading host intestinal cells. N. parisii initially develops in direct contact with the cytoplasm as a meront, eventually differentiating into a transmissible spore form that exits the cell[32]. Although the infection reduces worm lifespan, infected animals can generate enormous numbers of spores before death, with a single worm able to produce over 100,000 spores during the course of the infection[30,33]. Using C. elegans, we now report the first unbiased identification of microsporidia host-exposed proteins inside of an animal. These proteins are enriched for signal sequences and transmembrane domains, and they are rapidly evolving and tend to belong to unique large gene families. We also find that these species-specific large families are common throughout microsporidia. Using the properties we identified for the host-exposed proteins in Nematocida, we analysed 23 microsporidia genomes to predict potential host-exposed proteins, almost all were found to have no known molecular function. These results suggest that microsporidia use a set of lineage-specific, rapidly evolving proteins to interact with their hosts. This study provides a foundation for further functional characterization of host-exposed microsporidian proteins, and demonstrates the utility of proximity-labelling proteomic methods to broadly identify pathogen proteins localized within host cells.

## Results

**Identification of Nematocida host-exposed proteins.** To identify microsporidia proteins that come into contact with the intracellular host environment, we used the technique of spatially restricted enzymatic tagging[13]. This approach uses the enzyme APX to label proteins in the compartment where the enzyme is expressed with a biotin handle for subsequent purification (Fig. 1a). We generated strains of C. elegans expressing GFP-APX, either in the cytoplasm or in the nucleus of intestinal cells (Fig. 1b). We also generated a negative control strain that expresses GFP in the intestine, but without the APX protein (Supplementary Table 1).

First, we inoculated these transgenic animals with N. parisii spores, which led to the majority of animals being infected (Supplementary Fig. 1). These animals were then incubated for 44 h at 20 °C to allow for growth of the parasite. Next, we added the biotin-phenol substrate and hydrogen peroxide to these animals to facilitate APX-mediated biotinylation of host and pathogen proteins proximal to the GFP-APX protein. Under these conditions, we detected biotin-labelled proteins by microscopy in the intestinal cells of infected animals, but no labelling in the microsporidia cells themselves, demonstrating that the labelling technique is restricted to host-cell regions (Supplementary Fig. 2). Biotinylated proteins were isolated from total worm extracts using streptavidin-conjugated resin, and these purified proteins were identified using mass spectrometry. Biotinylated proteins from infected animals were isolated in triplicate and over 4,000 proteins from C. elegans and N parisii were identified (Supplementary Fig. 3).

As validation that proteins were labelled in specific compartments in this experiment, we used the labelled C. elegans proteins as an internal control. By comparing spectral counts

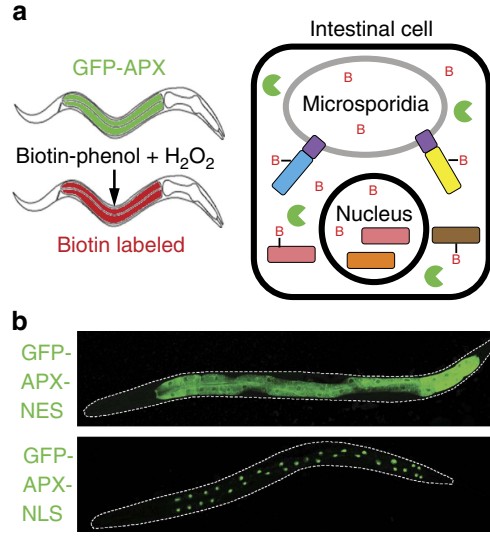

**Figure 1 | Overview of approach to detect and analyse host-exposed microsporidia proteins.** (**a**) Schematic of spatially restricted enzymatic tagging in *C. elegans*. Left, worms expressing GFP-APX in the cytoplasm of the intestine and infected with microsporidia are treated with biotin-phenol and $H_2O_2$. This treatment results in proteins within the intestinal cytoplasm being labelled with biotin. Right, an intestinal cell infected with microsporidia expressing cytoplasmic APX (green circular sectors) labelling microsporidia host-exposed proteins with biotin (red B). (**b**) Animals expressing GFP-APX in the intestine localized to either the cytoplasm (top) or the nucleus (bottom).

identified in the cytoplasmic APX, nuclear APX and no APX strains, we identified 891 *C. elegans* proteins specifically labelled in the intestine (Supplementary Data 1). By comparing *C. elegans* proteins in the cytoplasmic and nuclear samples, we identified 118 proteins specific to the nucleus and 114 proteins specific to the cytoplasm. We then compared these proteins to *C. elegans* proteins with previously reported localization. The set of proteins we identified as either cytoplasmic or nuclear specific are enriched for proteins known to be localized in that subcellular compartment (Supplementary Fig. 4A).

Comparing proteins from the cytoplasmic APX and nuclear APX samples to the no APX sample, we identified 72 *N. parisii* proteins that were enriched above background levels, as defined by the no APX strain (Supplementary Data 2). To approximate the total microsporidia proteome detectable in our experiments, we identified 392 *N. parisii* proteins from the no APX control samples (see 'Methods' section). We then compared these protein sets to previously generated RNAseq expression data[22]. The host-exposed proteins that we identified had moderate mRNA expression levels, with few detected from either the lowest or highest expressed mRNAs (Fig. 2a). In contrast, proteins identified in the no APX control strain are among the most highly expressed mRNAs in the genome (Supplementary Fig. 5A). This result suggests that the host-exposed proteins we identified are not biased towards highly expressed proteins.

Compared with all proteins in the genome, the host-exposed proteins we identified were significantly enriched in both signal peptides and transmembrane domains: over 75% of the proteins identified (enrichment *P*-value of $6.6E-13$) had at least one predicted targeting signal (Fig. 2b). Neither the proteins identified from the no APX control, nor the identified *C. elegans* intestinal proteins are enriched for these targeting signals compared with the genome (Supplementary Figs 4B and 5B). Altogether, the results indicate that our spatially restricted enzymatic tagging technique identified a high-quality data set of *N. parisii* host-exposed proteins in *C. elegans*.

If the host-exposed proteins we identified were truly secreted from *Nematocida* parasite cells into *C. elegans* host cells, we would predict that they would be processed by a signal peptidase in the parasite that would cleave off the signal sequence, and we could then detect the resulting N-terminal fragments in the host. Indeed, we detected N-terminal peptides corresponding to the predicted signal peptidase processed form for 4 of the 22 identified host-exposed proteins with signal peptides (two from *N. parisii* and two from *N.* sp. 1—see below), providing support that microsporidia host-exposed proteins containing signal peptides are secreted into the host (Supplementary Data 2 and 3 and Supplementary Table 2).

To investigate the subcellular localization of *N. parisii* host-exposed proteins, we compared proteins identified from animals expressing APX in the cytoplasm to those identified from animals expressing APX in the nucleus. From this comparison we found four proteins specific to the nucleus and eight proteins specific to the cytoplasm. Of the four nuclear specific proteins, three are predicted to have signal peptides, while all eight cytoplasmic specific proteins are predicted to have transmembrane domains. These data provide support for a model where proteins with signal peptides are secreted into the host cell and can localize to different cellular compartments, including the nucleus. Proteins containing transmembrane domains are likely attached to the membrane of the pathogen where they come in contact with the host cytoplasm (Fig. 2c).

**Many host-exposed proteins belong to large gene families**. Large, expanded gene families have been suggested to mediate host–pathogen interactions in a number of pathogen species and several large gene families have been previously identified in *Nematocida* species[22,31,34]. We defined large gene families as groups of homologous proteins with at least 10 members in one species that were enriched in signal peptides or transmembrane domains. We initially identified these families from paralogous orthogroups and then generated profile hidden Markov models to identify additional members in the genome.

There are four large *N. parisii* gene families that contain from 18 to 169 members. Two of these gene families, NemLGF1 and NemLGF5, encode signal peptides, and the other two gene families, NemLGF3 and NemLGF4, encode C-terminal transmembrane domains. The host-exposed proteins we identified are significantly enriched (*P*-value of $1.3E-16$) in these families and contain 35 members of these four genes families, with at least one host-exposed protein in each of the four families (Fig. 2b,c). The four nuclear specific proteins are members of the NemLGF1 or NemLGF5 gene family, whereas four of the cytoplasmic specific proteins with transmembrane domains belong to the NemLGF3 family (Supplementary Data 2).

**Nematocida host-exposed proteins are clade-specific**. To investigate how the repertoire of *N. parisii* host-exposed proteins is evolving, we explored whether the identified host-exposed proteins are conserved in three other *Nematocida* species. The earliest known diverging species of the genus is *N. displodere*, which proliferates well in the epidermis and muscle, but poorly in the intestine[31]. In contrast, the other *Nematocida* species are intestinal-specific[30]. Previously, the species known to be the most closely related to *N. parisii* was the intestinal-specific *N.* sp. 1 (strain ERTm2), which shares 68.3% average amino acid identity with *N. parisii*. To provide a more closely related species for comparison, we sequenced and assembled the genome of *Nematocida* strain ERTm5, an intestinal-specific strain that was isolated from a wild-caught *C. briggsae* in Hawaii[35]. This strain was previously described as a strain of *N. parisii* based on rRNA

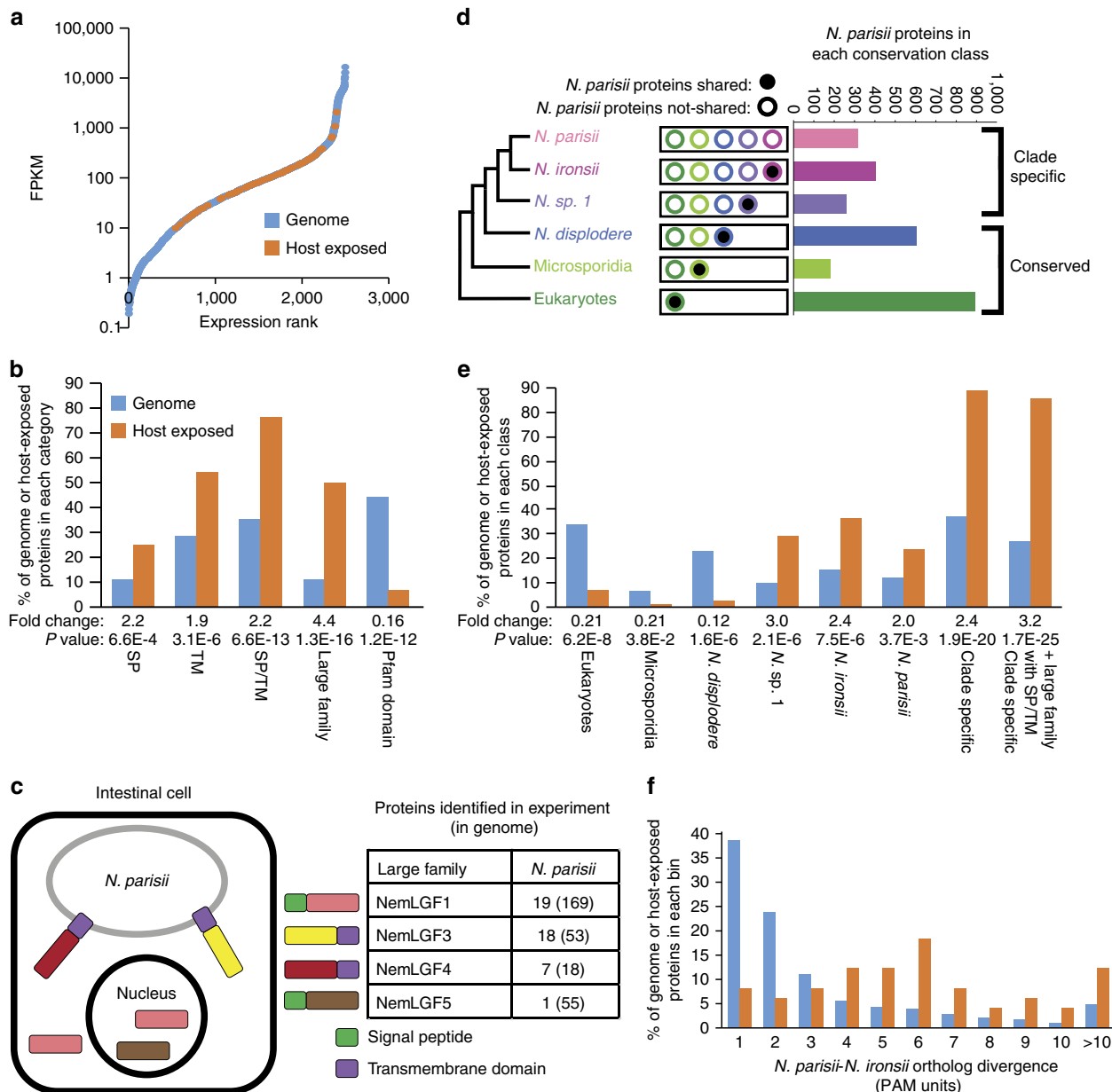

**Figure 2 | Properties of experimentally identified *N. parisii* host-exposed proteins.** (**a**) Comparison of mRNA expression levels of identified host-exposed proteins (orange dots) to the rest of the expressed *N. parisii* proteins (blue dots). Expression data are from a previous RNAseq study on animals infected for 30 h at 25 °C (ref. 58). (**b,e**). Comparison of identified host-exposed proteins (orange) to the genome (blue). Enrichment-fold change and *P*-values (one-side Fisher's exact test) of the host-exposed proteins compared with the genome are listed below each category. (**b**) Properties of 72 *N. parisii* host-exposed proteins. The percentage of the *N. parisii* genome and the percentage of the host-exposed proteins in each category are shown. TM, transmembrane. SP, signal peptide. (**c**) Left, model of where identified large gene family proteins are localized. Right, the number of proteins of each gene family identified as host exposed and the total number of gene family members present in the genome is shown in parentheses. (**d**) Schematic of the categorization of *N. parisii* proteins by conservation class. The 2,661 proteins in the genome were placed into six classes of decreasing conservation from proteins conserved with eukaryotes to being proteins unique to *N. parisii*. (**e**) Percentage of the genome and host-exposed proteins in each conservation class. (**f**) Distribution of protein sequence divergence between *N. parisii* and *N. ironsii* one-to-one orthologs. The genome contains 2,083 orthologs that met our criteria and the host-exposed proteins contain 49 orthologs (See 'Methods' section). The percentage of the identified host-exposed proteins (orange) and the genome (blue) is plotted. Wilcoxon two-sample test comparing sequence divergence of orthologs in the genome to the host-exposed proteins has *P*-value of 6.8E − 11.

sequence, but based on our analysis, it now appears to define a new species (see 'Methods' section). This genome is comparable in quality to other sequenced genomes as judged both by assembly statistics and the presence of proteins conserved throughout microsporidia (Supplementary Table 3). This new species, *Nematocida ironsii*, now represents the closest known sister species to *N. parisii* and has an average amino acid identity of 84.7% compared with *N. parisii* (Supplementary Fig. 6 and Supplementary Table 5). To examine conservation, each *N. parisii* protein was placed into an orthogroup using six eukaryotic and 23 microsporidian genomes. Every *N. parisii* protein was categorized into one of six classes of decreasing conservation: (1) *N. parisii* proteins conserved with other non-microsporidia eukaryotes, (2) conserved with other

microsporidia, (3) conserved with *N. displodere*, (4) conserved with *N.* sp. 1, (5) conserved with *N. ironsii* and 6) those that are unique to *N. parisii* (Fig. 2d).

Using this evolutionary approach, we found that the set of host-exposed proteins we identified are significantly enriched ($P$-value of 1.9E − 20) for less conserved proteins, with only 12% having orthologs outside of a group of closely related *Nematocida* species (*N.* sp. 1, *N. ironsii* and *N. parisii*, which we refer to as 'clade-specific'). In contrast, 63% of all *N. parisii* proteins in the genome have orthologs outside of this clade of *Nematocida* species (Fig. 2e). Most of these identified proteins don't have a predicated molecular function, with only five of these 72 proteins containing a predicted Pfam domain (Fig. 2b). To determine the rate of protein evolution, we calculated the protein sequence divergence between orthologous *N. parisii* and *N. ironsii* proteins. We found that the host-exposed proteins are rapidly evolving compared with the other proteins in the genome (Fig. 2f).

To examine whether the properties of the host-exposed proteins we identified were conserved in other microsporidia species, we performed spatially restricted enzymatic tagging on *C. elegans* infected with *N.* sp. 1. Although we identified fewer *C. elegans* and microsporidia proteins from *N.* sp. 1 infected animals, we nonetheless found ten proteins enriched over background (Supplementary Fig. 3 and Supplementary Data 3). These proteins have similar properties to those identified for *N. parisii* as they are enriched in targeting signals and clade-specific proteins (that is, proteins not conserved in other eukaryotes, microsporidia or *N. displodere*) (Supplementary Fig. 7). They also are enriched for being members of large gene families, including three members of NemLGF1 and one member of the *N.* sp. 1-specific family NemLGF6. We also identified two pairs of orthologs from the two species: hexokinase (NEPG_02043 and NERG_02003) and a NemLGF1 family member (NEPG_02370 and NERG_01049). To expand this analysis to a different microsporidia genus, we examined data previously generated from germinated *Spraguea lophii* spores. We found that proteins identified as secreted from these germinated spores were also enriched in the properties of signal peptides and clade-specific proteins (Supplementary Fig. 8)[29].

Overall, we find that host-exposed proteins are highly enriched in three properties: (1) they have targeting signals (signal peptides or transmembrane domains), (2) they belong to large gene families and (3) they are clade-specific. In fact, 85% of *N. parisii* host-exposed proteins identified are either members of large gene families or are clade-specific proteins with a signal peptide or transmembrane domain (enrichment $P$-value of 1.7E − 25) (Fig. 2e). Although the number of proteins we identified with these properties is 61, the total number of proteins with these properties encoded by the genome is 713.

Current limitations of proteomic methods suggest that this approach will not result in the complete identification of all host-exposed microsporidia proteins. To estimate the sensitivity of this method we compared the identified *C. elegans* intestinal proteins to the total number of mRNAs expressed in the intestine[36]. We also compared the total number of detected *N. parisii* proteins to the number encoded by the proteome. From these comparisons, we estimate that we identified between ∼ 8 and 24% of potential host-exposed proteins. This would mean that the total host-exposed proteome encoded by *N. parisii* is in the order of 300–900 proteins, a range that encompasses the number of proteins in the genome that have the properties enriched in the experimentally identified host-exposed proteins.

**Large families display lineage-specific expansions**. If most members of *N. parisii* large gene families are involved in

host–pathogen interactions, we would predict that they would also be rapidly evolving with species-specific radiations. The four large gene families of *N. parisii* contain a total of 295 members. Members of these four families are also present in the other species in this clade, *N.* sp. 1 and *N. ironsii*, but not any other microsporidia species (Figs 3a and 4). Phylogenetic trees of these families show expansion of family members specific to each species (Fig. 3b,c). Members from these families are often not conserved between species, with only 5–39% of *N. parisii* members in each gene family that have orthologs in *N.* sp. 1 and 56–95% that have orthologs in *N. ironsii* (Fig. 3c). The largest families that have signal peptides, NemLGF1 and NemLGF5, are enriched for genes on the ends of chromosomes, a chromosomal localization that is not enriched in the transmembrane-containing families (Supplementary Fig. 9A). The four families are often adjacent to each other, suggesting they are being generated through local duplication events (Supplementary Fig. 9B).

**Large families are common in microsporidia**. To examine whether large gene families are common in other microsporidia species, we examined 23 microsporidia genomes (17 other microsporidia species and six from *Nematocida*) (Supplementary Fig. 6). From these 21 species, 68 families were identified with at least 10 members in one species and enriched in either predicted signal peptides or transmembrane domains. In addition, we found that most (59 of 68) of these families do not have any members present outside of the genus or species. For example, there are three families with members present in all four *Encephalitozoon* species but no other species examined. In addition, we identified four large gene families that were conserved throughout most microsporidia including two ricin B domain-containing families[29]. All but one species examined has a large genus-specific family, demonstrating that large gene families are widespread throughout microsporidia.

**Prediction of host-exposed proteins from other microsporidia**. We next investigated whether proteins that are not widely conserved in microsporidia share properties with the identified host-exposed proteins. We examined 23 microsporidian genomes to identify proteins that are not conserved with other eukaryotes, or conserved with distantly related microsporidia species. These clade-specific proteins are all significantly enriched in targeting signals compared with proteins conserved with more distally related microsporidia or other eukaryotes (Fig. 5a). This result is similar to what we found in our analysis of experimentally identified host-exposed proteins in *Nematocida*, and similar to a previous study of several microsporidian species[37].

Our analyses above indicated that the genomes of microsporidia contain two classes of proteins enriched in targeting signals, clade-specific proteins and large gene families. Most of the proteins (85%) we identified experimentally in *N. parisii* also display these characteristics. On the basis of these genomic signatures and our experimental results, putative host-exposed proteins for each species were predicted. These predictions of 11,675 proteins for 23 genomes are provided as a resource in Supplementary Data 5. Although these characteristics alone may not be sufficient to direct proteins to become host exposed, these proteins likely represent a substantial portion of the host-exposed proteins that each species uses and provide an unprecedented set of candidates for future studies.

The potential host-exposed proteins account for 6–32% of the genome of each species. Interestingly, the number of predicted host-exposed proteins can vary even within closely related species, with *E. cuniculi* having almost twice as many predicted proteins as the other members of the genus (Fig. 5b). The majority of these

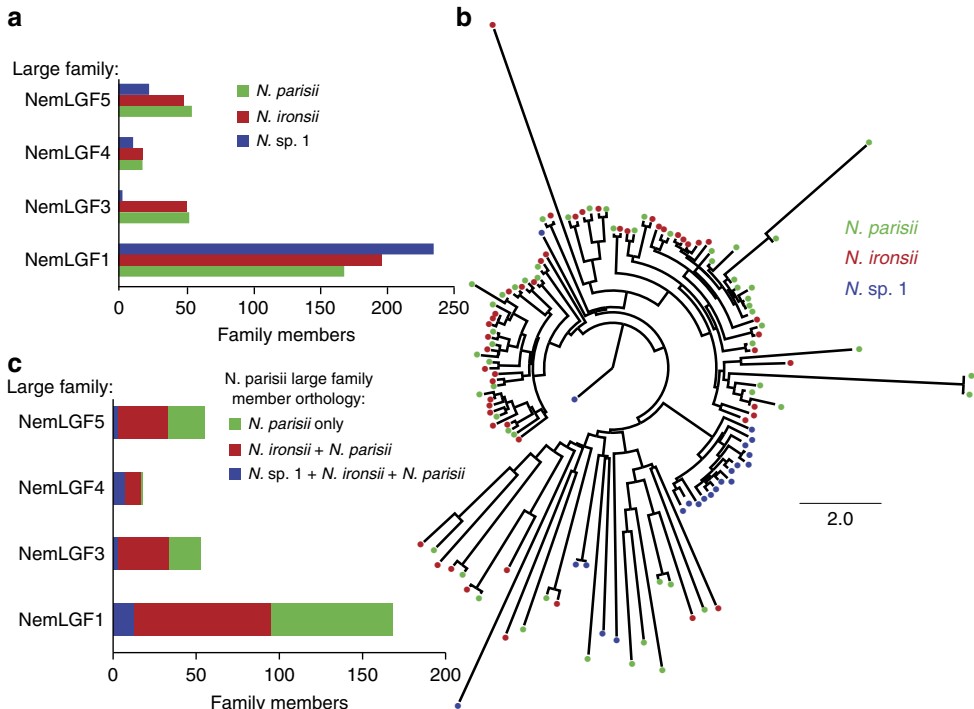

**Figure 3 | Species-specific radiation of large gene family members. (a)** Number of members for each gene family that are present in each species. **(b)** NemLGF5 tree showing gene family-specific radiation. Scale is changes per site. **(c)** The number of proteins for the indicated gene family that are either unique to *N. parisii* (green), have orthologs in *N. ironsii* (red) or have orthologs in *N. sp. 1* and *N. ironsii* (blue).

putative host-exposed proteins do not have a predicted molecular function, with only 7.4% having a predicted Pfam domain that occurs in proteins outside of microsporidia (Supplementary Data 5). Although most of these proteins do not have known domains, several species have expanded families of leucine-rich repeat (LRR) domains and two species have expanded families of protein kinases (Fig. 5). The most frequently observed domains in putative host-exposed proteins that are not members of the large gene families are transporters, kinases, LRR domains, ubiquitin carboxyl-terminal hydrolases and the bacterial specific DUF1510 (Supplementary Fig. 10)[38]. Interestingly, a number of domains that are present in the large gene families are also observed in the non-paralogous proteins, suggesting that there are several common domains that have been utilized in multiple microsporidia species to interact with hosts. These predictions of host-exposed proteins suggest that microsporidia employ a large number of proteins with novel domains to interact with hosts.

## Discussion

To understand how microsporidia interact with their hosts, we experimentally identified 82 host-exposed proteins from two *Nematocida* species. To identify these proteins, we employed an unbiased approach that labelled the host-exposed pathogen proteins inside of an intact animal. Interestingly, of the identified proteins, we found that four proteins were specific to the nucleus, where they may affect host transcription or nuclear organization, as has been shown for effectors from other pathogens[5,39]. Attempts to validate these host-exposed proteins using orthogonal experimental approaches have not been possible due to the lack of specific antibodies against *Nematocida* proteins and the lack of genomic modification techniques for microsporidia[40]. Nonetheless, this approach was able to identify *C. elegans* proteins previously shown to be localized to the nucleus and cytoplasm, validating the specificity of the technique. This approach of

tagging pathogen proteins based on their localization is likely to be useful in the study other *C. elegans* pathogens as well as a general tool to examine putative pathogen effector proteins in a range of hosts[41,42].

A key feature of the identified host-exposed proteins is their enrichment in signal peptides and transmembrane domains. This enrichment suggests that these are the two major targeting signals that are used in *Nematocida* for proteins to become exposed to the host, as they are present in 76% of identified proteins. Such signals might be missed in the remaining proteins due to the lack of sensitivity of these prediction methods and the misannotation of the true N and C termini of *Nematocida* proteins[19,20]. The identified proteins could also be useful to discover potential secondary signals in the proteins that direct transmembrane and signal peptide-containing proteins to become host exposed, rather than to other membranes inside microsporidia[43].

We found that large gene families are common within microsporidia, with 68 gene families from 23 microsporidia genomes being identified. Although several of these families had been previously reported, here we provide a comprehensive identification of these gene families throughout microsporidia[29,31,44,45]. The majority of these large gene families have no known molecular function based on sequence similarity. One enticing possibility is that the expansion of these families is due to interactions with host proteins. In support of this possibility, a number of the gene families with predicted domains are known to mediate protein–protein interactions including LRR and RING domains.

One intriguing characteristic of these large gene families is that they are either genus- or species-specific, with large lineage-specific expansions of these gene families across microsporidia. The differences in the total number of gene families can be quite large in the same genus. For example, in the family NemLGF3, *N. sp. 1* only has three members compared with 53 members in *N. parisii*. Both strains of *N. sp. 1* also have a gene family

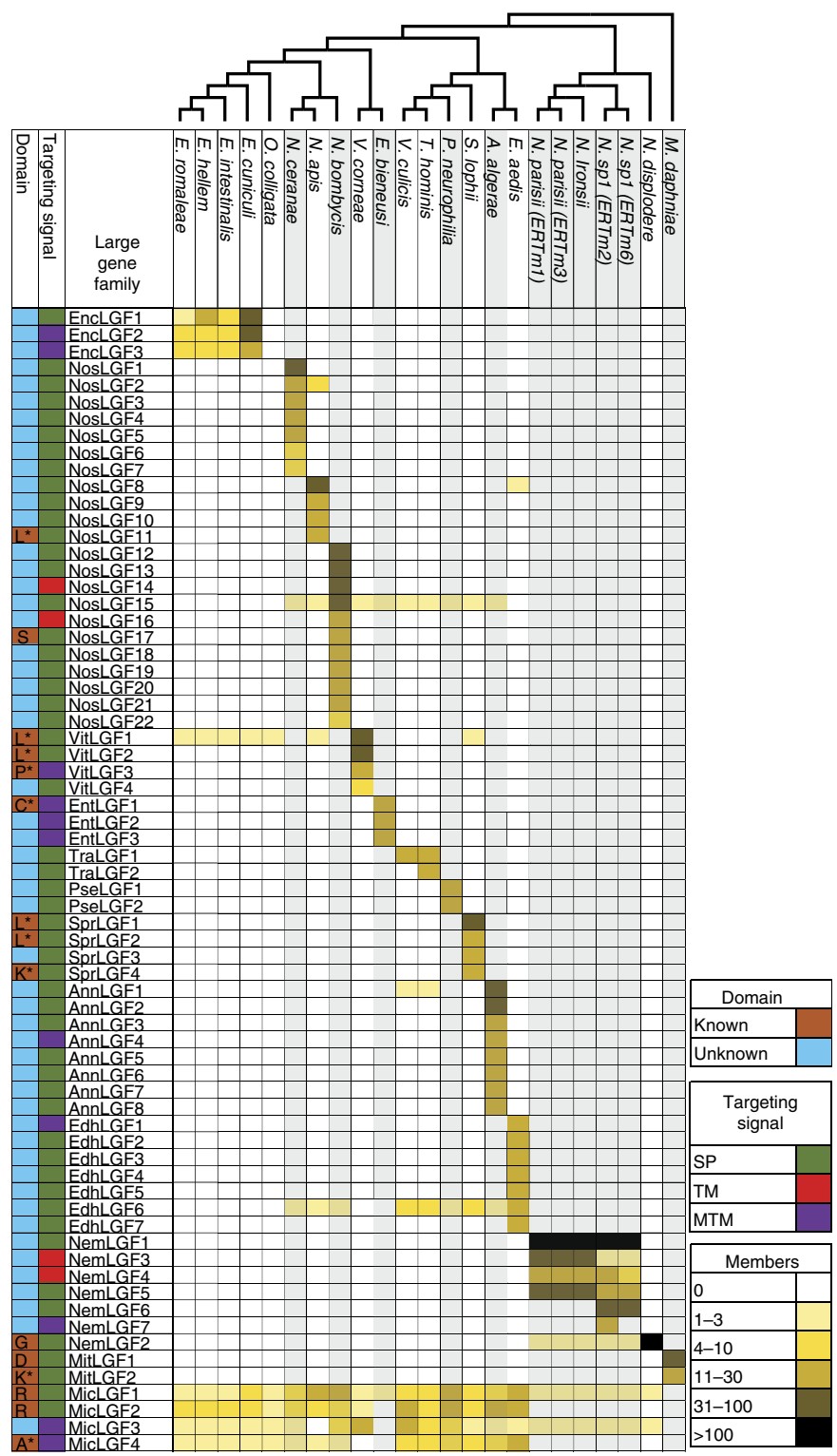

**Figure 4 | Large gene families are widespread throughout microsporidia.** Heat map showing large gene families identified in microsporidia and the number of gene family members in each species. Cladogram of species is shown at the top. Each column represents a species and strains are shown in parentheses. Each clade of species is alternatively shaded in grey or white. Each row represents a large gene family. Families are named and clustered based on the genus from where they were identified. The first column indicates if a known Pfam domain can be found within the indicated large gene family. Domains defined as follows: L (LRR), S (serpin), P (peptidase M48), C (chitin synthase), K (kinase), D (Duf3638), R (RicinB) and A (ABC transporter). Members of each gene family were determined using HMMER, except for those indicated with an *, which were determined using OrthoMCL. The second column indicates the targeting signal that is overrepresented within the indicated gene family. SP, signal peptide, TM, single transmembrane domain and MTM, multitransmembrane domain. Each box in columns to the right of the gene family name is coloured according to the total number of members within a given gene family.

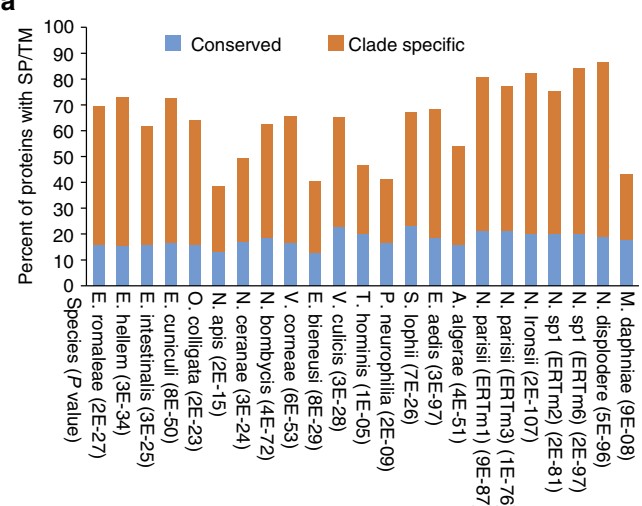

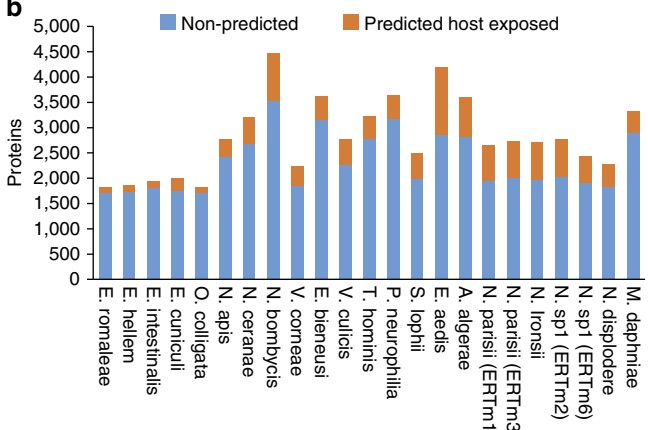

**Figure 5 | Prediction of host-exposed microsporidia proteins.** (**a**) Clade-specific microsporidia proteins (orange) are enriched in signal peptides/transmembrane domains compared with conserved proteins (blue). Enrichment *P*-values (one-sided Fisher's exact test) are listed in parenthesis below each species. (**b**) The number of proteins in each microsporidia genome that are predicted to be host-exposed proteins (orange), compared with the rest of the genome (blue).

(NemLGF6) that is absent in other microsporidia, but the ERTm2 strain of *N*. sp. 1 has 23 members of a multitransmembrane gene family (NemLGF7) that is absent from the ERTm6 strain (Fig. 4). These differences suggest that the composition and emergence of these gene families can change rapidly.

Several of the large gene families in *Nematocida* contain over 100 members and constitute a sizable portion of the entire genome. For example, members of NemLGF1 account for 6.4% of the genome of *N. parisii* and members of NemLGF2 account for 10.8% of the *N. displodere* genome. The exact forces that are providing pressure for gene family expansion in microsporidia are unclear, though one likely possibility is that variation in the host environment shapes the expansion of pathogen protein families. In the case of *N. displodere*, the pathogen has been observed to replicate in multiple tissues, and this variation in cellular environment could drive family diversification. Another possibility is that the genetic diversity of the hosts being encountered could drive the expansion. The complete native ecology of hosts that *Nematocida* interact with is unknown, though both *C. elegans* and *C. briggsae* have been found infected with *Nematocida* microsporidia[30]. For other

microsporidia species there is both ecological evidence and laboratory studies demonstrating that the same strain of microsporidia can infect closely related host species[15,46,47]. We speculate this host diversity could drive the expansion of large gene families in microsporidia and that these large gene families may in turn influence the host range.

The majority of the host-exposed proteins we identified in *N. parisii* and *N*. sp. 1 were proteins not conserved with *N. displodere* or other microsporidia species. Although lack of conservation accounts for most of the proteins identified, several conserved proteins were identified, including hexokinase, which we identified in both *Nematocida* species. Hexokinase was previously found to have predicted signal peptides in several microsporidia species and to be secreted in a heterologous system, and to be secreted from the microsporidia *Antonospora locustae*, providing experimental evidence that secreted hexokinase is a conserved feature of microsporidia[22,28,37]. There are also several large gene families that have members present in multiple microsporidia species. This observation suggests that although selective forces result in a host-exposed protein repertoire with many unique proteins for each microsporidia clade, there are some proteins conserved throughout microsporidia involved in host interactions.

A number of forces are likely to shape the repertoire of host-exposed proteins, including the selective pressure of the host and interactions with other pathogens. Many of the features of the host-exposed protein repertoire in microsporidia are similar to characteristics reported in the apicomplexan phylum of protozoan obligate intracellular pathogens. Large gene families with either signal peptides or transmembrane domains are common. These families often have subtelomeric genomic locations and are species specific[34]. Over 200 secreted proteins have been predicted in *P. falciparum* and few are conserved with other *Plasmodium* species[3]. Most of these proteins also have no predicted molecular function[43]. These similarities among species suggest that similar selective pressures can sculpt a host-exposed protein repertoire with related properties. In contrast, strains of the bacterium *P. syringae* are predicted to have less than 40 type III effectors, many of which are shared with other bacteria and display evidence of horizontal gene transfer[9,48].

A striking result of our analysis is that a large number of experimentally identified and predicted host-exposed proteins do not have domains found outside of microsporidia. These host-exposed proteins are a potential source of novel biochemical activity as the extreme selective pressures inflicted on pathogens by the host has been shown to result in unique molecular functions[49,50]. Interestingly, we also predict a large per cent of the microsporidia genome to be responsible for mediating host–pathogen interactions. This suggests that although microsporidia have the smallest known genomes of any eukaryotes, they somewhat paradoxically encode a substantial cadre of proteins for interacting with their hosts. Understanding how microsporidia use these proteins to mediate host interactions will provide insight into their impact on hosts and the constraints on evolution of a minimalistic eukaryotic genome.

## Methods

**Cloning and generation of *C. elegans* expressing APX.** Soybean APX (W41F) was optimized for *C. elegans* expression using DNAworks to design primers[51]. These primers were annealed using a two-step PCR method and cloned into Gateway plasmid pDONR 221. Gibson cloning was then used to introduce GFP as an N-terminal fusion, and NES (LQLPPLERLTLD) and NLS (PKKKRKVD PKKKRKVDPKKKRKV) tags to the C-terminus of APX[52]. One kilobase (kb) of sequence upstream of the intestinal-specific gene *spp-5* was used as a promoter and *unc-54* as a 3′ sequence. Multisite Gateway was used to combine these fragments into the plasmid pCFJ150 to generate targeting constructs. The MosSCI approach was used to generate single copy insertions by injecting *unc-119* mutants from the EG6699 strain with these targeting constructs[53]. Each transgenic strain was

backcrossed to the wild-type N2 strain three times and the homozygote was used in subsequent experiments.

**Spatially restricted enzymatic tagging in C. elegans.** C. elegans strains that express GFP-APX either localized to the cytoplasm or nucleus, as well as a control GFP only strain were used (Supplementary Table 1). Mixed-stage populations of each strain were grown at 20 °C on nematode growth media (NGM) plates seeded with OP50-1 bacteria. Animals were washed off of plates with M9 and treated with sodium hypochlorite solution/1 M NaOH for 2–3 min. Eggs were washed three times with M9 and resuspended in 5 ml of M9 in a 15 ml tube. These eggs were incubated 18–24 h at 20 °C on a rotator to hatch L1 animals. Animals were infected with microsporidia, N. parisii (strain ERTm1) and N. sp. 1 (strain ERTm2), using spores that were purified as previously described[30]. Infections were performed in 15 ml tubes containing ∼150,000 L1s in 500 µl M9 and 10 µl of 10× concentrated OP50 bacteria, to which 405 µl of N. parisii ($44.45 \times 10^6$ spores) or N. sp. 1 ($14.85 \times 10^6$ spores) were added. These animals were incubated with spores for 4 h at 20 °C. Animals were then washed three times with M9. Animals were resuspended in 12.5 ml M9 and 2.5 ml was added to each 15 cm RNAi plates seeded with HT115 bacteria expressing bus-8 RNAi feeding clone[54]. This RNAi clone increases permeability of the cuticle and allows for efficient biotin labelling[55]. Infected animals were grown for 44 h at 20 °C. Animals were recovered off of each plate with M9T (M9/0.1% Tween-20) and animals washed once more with M9T. To worms in a total of 100 µl M9T in 1.5 ml tubes, 900 µl of labelling solution (0.1% Tween-20, M9, 3.3 mM biotin-phenol, synthesized as previously described[13]) was added. Worms were incubated for 1 h at 22–24 °C on a rotator. Then 10 µl of 100 mM $H_2O_2$ was added for 2 min. The reaction was quenched with 500 µl quench buffer (M9/0.1% TWEEN-20/10 mM sodium azide/10 mM sodium ascorbate/5 mM Trolox). Samples were washed four times with 1 ml with quench buffer. To each worm pellet 800 µl lysis buffer (150 mM NaCl/50 mM TRIS pH 8/1% TritonX-100/0.5% sodium deoxycholate/0.1% SDS/10 mM sodium azide/protease complete tablet (Roche)/10 mM sodium ascorbate/5 mM Trolox/1 mM PMSF) was added and worms were then immediately frozen dropwise in liquid $N_2$.

Frozen worm pellets were ground to a fine powder in liquid $N_2$ to generate protein extracts. These protein extracts were then centrifuged for 10 min 21,000g at 4 °C. The supernatant was then filtered over a desalting column (Pierce). The protein concentrations of the extracts were normalized using a Pierce 660 nm Protein Assay. To 340 µg of each sample was added 25 µl of high-capacity streptavidin agarose resin (Pierce) in a total of 700 µl lysis buffer. Extracts were incubated with beads for 1 h on rotator. Beads were then washed five times with 1 ml lysis buffer, three times with 1 ml 8 M urea/10 mM TRIS pH 8 and three times with 1 ml PBS. The liquid was removed from the beads and 100 µl of 0.1 µg µl$^{-1}$ trypsin (Promega)/50 mM NaHCO$_3$ was added to each sample and incubated at 37 °C for 24 h.

**LC–MS–MS parameters.** Samples were analysed in triplicate by LC–MS/MS using a Q-Exactive mass spectrometer (Thermo Scientific, San Jose, CA, USA) with the following conditions. The following is a generalized nHPLC and instrument method that is representative of individual analyses. Peptides were first separated by reverse-phase chromatography using a fused silica microcapillary column (100 µm ID, 18 cm) packed with C18 reverse-phase resin using an in-line nano-flow EASY-nLC 1,000 UHPLC (Thermo Scientific). Peptides were eluted over a 100 min 2–30% ACN gradient, followed by a 5 min 30–60% ACN gradient, a 5 min 60–95% gradient, with a final 10 min isocratic step at 0% ACN for a total run time of 120 min at a flow rate of 250 nl min$^{-1}$. All gradient mobile phases contained 0.1% formic acid. MS/MS data were collected in a data-dependent fashion using a top 10 method with a full MS mass range from 400 to 1,800 $m/z$, 70,000 resolution, and an AGC target of 3e6. MS2 scans were triggered when an ion intensity threshold of 1e5 was reached with a maximum injection time of 60 ms. Peptides were fragmented using a normalized collision energy setting of 25. A dynamic exclusion time of 40 s was used and the peptide match setting was disabled. Singly charged ions, charge states above 8 and unassigned charge states were excluded.

**Peptide and protein identification and quantification.** The resultant RAW files were converted into mzXML format using the ReadW.exe programme. The SEQUEST search algorithm (version 28) was used to search MS/MS spectra against a concatenated target-decoy database comprised of forward and reversed sequences from the reviewed UniprotKB/Swiss-Prot FASTA C. elegans database combined with the UniprotKB E. coli (K12 strain) database, and the N. parisii and N. sp. 1 predicted proteomes with common contaminants appended. The search parameters used are as follows: 20 parts per million (p.p.m.) precursor ion tolerance and 0.01 Da fragment ion tolerance; up to three missed cleavages were allowed; dynamic modification of 15.99491 Da on methionine (oxidation). Peptide matches were filtered to a peptide false discovery rate of 2% using the linear discriminant analysis[56]. Proteins were then filtered to a 2% false discovery rate (FDR), which resulted in a peptide FDR below 1%. Peptides were assembled into proteins using maximum parsimony and only unique and razor peptides were retained for subsequent analysis. Peptide spectral count data was mapped onto the assembled proteins and used for subsequent analysis.

**Analysis of mass spectrometry data.** The peptide spectral counts of proteins were used to calculate fold change ratios and FDR P-values between GFP only, NES and NLS samples using the qspec-param programme of qprot_v1.3.3 (ref. 57). Several criteria were used to classify proteins as being host-exposed proteins; no counts in the GFP only samples and an average greater than 2 peptides in the NES samples or an NES/GFP ratio greater than 2-fold with an FDR P-value of <0.005. In addition proteins with an NLS/GFP ratio of greater than 3-fold were included. Proteins were classified as being NLS-enriched if they had a greater than a 2-fold NLS/NES ratio and NLS depleted if they had greater than a four-fold NES/NLS ratio. All data for N. parisii proteins is in Supplementary Table 3 and for N. sp. 1 proteins in Supplementary Table 6. C. elegans intestinal proteins were detected in the same way as described above and data are in Supplementary Table 2. N. parisii proteins in the no APX sample were required to have an average of greater than two peptides in the GFP only sample.

**Microscopy of infected C. elegans.** To detect biotin labelling in infected worms, intestines were dissected and stained with anti-GFP (Roche) and Streptavidin Alexafluor 568 (Thermo Fisher). Images were taken using a Zeiss LSM700 confocal microscope with a ×40 objective. To detect microsporidia in infected worms, fluorescence in situ hybridization with probes specific for microsporidia was performed as previously described and imaged with a Zeiss AxioImager M1 microscope[58].

**Genome sequencing and analysis.** Genomic DNA was obtained from Nematocida strain ERTm5 infected C. elegans by phenol-chloroform extraction, treated with RNAse for 1 h, and then precipitated with ethanol and resuspended in TE buffer. One lane of 100 bp paired-end sequencing on an Illumina HiSeq 2000 (Cofactor Genomics) was used to generate reads which were filtered to remove C. elegans and E. coli genome reads.

The genome was assembled and annotated as done previously[31]. Although ERTm5 was previously considered to be a strain of N. parisii based on 100% nucleotide identity of 18S ribosomal RNA sequences[35], the average nucleotide identity across the genome between N. parisii strain ERTm1 and ERTm5 is 92.3%, which is calculated using the nucmer programme in mummer 3.23 (ref. 59). The two strains are more dissimilar than the generally accepted definition of different microbial species having less than 95% average nucleotide identity[60]. Because of this, we consider strain ERTm5 to be a Nematocida species distinct from N. parisii. Because the strain ERTm5 was isolated from C. briggsae found in Kauai, Hawaii, we named this new species N. ironsii in dedication to the Hawaiian surfer Andy Irons. Assembly statistics for all microsporidia species used in this study are in Supplementary Table 4. Annotation of N. ironsii proteins are in Supplementary Table 5. Conservation of proteins for each microsporidia species was determined by counting the number of orthogroups conserved between all 23 genomes divided by the number of orthogroups present in the other species. A phylogenetic tree of the microsporidia species was generated as described previously (Supplementary Fig. 6)[31].

**Functional annotation of microsporidia proteins.** Domains were predicted with the Pfam-A 28.0 library using the HMMscan function in HMMER 3.1 with an E-value cutoff of $<10^{-5}$. Prediction of signal peptides was done using SignalP 4.1, using the best model with a cutoff of 0.34 for both the noTM model and for the TM model[19]. These same cutoffs were used to generate signal peptide processed forms. Prediction of transmembrane domains was done using TMHMM 2.0 (ref. 20).

**Determination of microsporidia orthogroups.** Conservation of proteins was determined using OrthoMCL 2.0.9 (ref. 61). This analysis was performed using six eukaryotic genomes (Saccharomyces cerevisiae, Monosiga brevicollis, Rozella allomycis, Neurospora crassa, Ustilago maydis and Allomyces macrogynus) and 23 microsporidia genomes (Supplementary Table 4) using an inflation index of 1.5 and a BLAST E-value cutoff of $10^{-5}$.

**Identification of large gene families in microsporidia.** Families were initially identified from microsporidia orthogroups. Proteins in each initial group were aligned using MUSCLE 3.8.31 (ref. 62), and profile HMM models were built using HMMbuild. The microsporidia genomes were then searched using HMMscan with an E-value cutoff of $10^{-5}$. This process was performed iteratively until no more additional proteins met the cutoff. The following domains are widely present in eukaryotic species so family membership was determined using orthogroups: LRR, kinase, ABC transporter, peptidase and chitin synthase. To be considered a large gene family, at least 10 proteins had to belong to the family in a single microsporidia genome. In addition, each family was required to have at least 2-fold enrichment over the genome in either predicted signal peptides or transmembrane domains. Families were named by first three letters of genus and numbered based on size. Those that were present in multiple genera were named with the prefix 'Mic'.

**Determination of N. parisii large gene family orthologs.** For each of four large gene families (NemLGF1 and NemLGF2-4) members from N. parisii (ERTm1 and ERTm3), N. ironsii and N. sp. 1 (ERTm2 and ERTm6) were aligned using MUSCLE. Phylogenetic trees were inferred for each family using RAxML 8.2.4

(ref. 63) using the PROTGAMMALG model and 1,000 bootstrap replicates. For NemLGF1, an initial tree was generated using 10 bootstrap replicates and then divided into seven sub trees. Orthologs of *N. parisii* proteins in each family were manually assigned using these maximum likelihood trees. To determine the genomic location of these families the 5 largest scaffolds of *N. parisii* (ERTm1) were used. Chromosomal ends were defined as the first and last 30 kb of each scaffold. Adjacent proteins were calculated as where the next protein was next to it.

**Determination of conservation.** For *N. parisii* proteins, conservation was determined based on orthogroups, except for the large gene families NemLGF1 and NemLGF2-4 for which orthology was determined as described above. The following procedure was used to place the *N. parisii* proteins into six categories. If a *N. parisii* protein was in any group with a protein from the six non-microsporidian eukaryotic species, the protein was placed in the category 'Eukaryotes'. If any remaining unassigned proteins were in a group with a protein from the microsporidia species not in the genus *Nematocida*, then it was placed in the category 'microsporidia'. If any remaining unassigned proteins were in a group with an *N. displodere* protein, then it was placed in the category '*N. displodere*'. If any remaining unassigned proteins were in a group with an *N. sp. 1* protein, then it was placed in the category '*N. sp. 1*'. If any remaining unassigned proteins were in a group with a *N. ironsii* protein, it was placed in the category '*N. ironsii*'. The remaining proteins were placed in the category '*N. parisii*'.

To predict host-exposed proteins the conservation of microsporidia proteins was determined. Proteins of each species were placed into two classes, 'Conserved' or 'clade-specific'. If a protein was in the same group as a protein from any of the eukaryotic or microsporidia species then it was classified as 'conserved'. Otherwise it was classified as 'clade-specific'. This was done except for the closer-related species, where proteins in the same clade were not considered. For this purpose the following clade definitions were used: *Nematocida* species are *N. parisii*, *N. sp. 1*, and *N. ironsii*; *Encephalitozoon* species are *E. romaleae*, *E. hellem*, *E. intestinalis*, *E. cuniculi* and *O. colligate*; and the species *V. culicis* and *T. hominis*.

**Calculation of protein sequence divergence.** Proteins for microsporidia genomes were placed into orthogroups as described above. Proteins from one-to-one orthologs of the two *N. parisii* strains (ERTm1 and ERTm3) and *N. ironsii* were aligned using MUSCLE 3.8.31 (ref. 62). For large gene families orthologs were determined as described above. For proteins conserved with *N. sp. 1*, the evolution rate was only calculated for one-to-one orthologs between the five genomes. For proteins conserved with *N. displodere*, the evolution rate was only calculated for one-to-one orthologs between all six *Nematocida* genomes. Maximum likelihood trees were built using ortholog sets (three sequences per set) of aligned protein sequences using PHYLIP (http://evolution.genetics.washington.edu/phylip.html). The sum of the sequence tree length dived by the number of sequences, in PAM units, was calculated for each ortholog set.

**Data availability.** The Whole Genome Shotgun project for the *N. ironsii* genome has been deposited at DDBJ/ENA/GenBank under the accession LTDK00000000. The version described in this paper is version LTDK01000000. All data supporting this manuscript is available from the corresponding author upon request.

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

## Acknowledgements

We thank Steven Wasserman, Matthew Daugherty, Robert Luallen, Kirthi Reddy, Michael Botts, and Lianne Cohen for providing helpful comments on the manuscript. A.W.R is a Monsanto Fellow of the Life Sciences Research Foundation. Some *C. elegans* strains were provided by the *Caenorhabditis* Genetics Center, which is funded by National Institutes of Health (NIH) Office of Research Infrastructure Programs Grant P40 OD010440. This work was supported by a New Scholar in Aging award from the Ellison Medical Foundation (E.J.B.) and R01GM114139, the David and Lucile Packard Foundation and a Burroughs Wellcome Fund fellowship (E.R.T.).

## Author contributions

A.W.R. designed, conducted and analysed experiments. K.M.B. provided the N. *ironsii* genome sequence. E.J.B. performed the mass spectrometry analysis. E.R.T. provided mentorship and with A.W.R and E.J.B. prepared the manuscript.

## Additional information

**Competing financial interests:** The authors declare no competing financial interests.

