## [Peer Review File · Nature Communications]

Reviewers' comments:

Reviewer #1 (Remarks to the Author):

In this manuscript the authors adapted for *C. elegans* an elegant and powerful methodology to differentially tag and identify proteins through proteomics that are located either in the nucleus or the cytosol of the nematode and exploited this approach to identify effector proteins from the microsporidia Nematocida, a natural intracellular pathogen of the nematode. The authors complement their proteomics MS analyses with a number of detailed bioinformatics analyses to further characterise the properties of the proteins of interest and ensure these are indeed strong candidate effector proteins of two Nematocida species. Other new data include the genome sequence of a new species of Nematocida and a global bioinformatics analysis of gene families across available annotated microsporidia genomes. The different claims of the authors are strongly supported by these very well presented data.

This is a very important study, which also represent a technical tour de force, as very little is currently know about effector proteins mediating host-microsporidia interactions and these intracellular pathogens are increasingly recognised as important in animals and humans in a number of ecosystems and medical contexts. More generally this methodology will undoubtedly influence the study of numerous host-intracellular pathogens systems. The paper is very nicely put together presenting experiments carefully performed and data conservatively interpreted. I have only a few details that should be considered to slightly improve the presentation and interpretations of these data and boost the "selling" of the used methodology to study host-intracellular pathogens interactions.

1) A key feature of a subset of the identified proteins is the presence of an N-terminal signal peptide (SP). Is there any evidence from the MS data, for some proteins at least, that the SPs were processed by the RER signal peptidase (SPase), assuming that a SPase is present in Nematocida? This would nicely link the localisation data with the molecular and cellular processes underlying the secretion of these proteins by the pathogen.

2) As stated in the manuscript only few microsporidia proteins mediating interactions with their host have been identified. I thought that here the authors could have been more scholarly by citing some additional papers describing these. Currently only two, relevant, papers are cited (23,24). Indeed the presented data, some in the supplementary material, refer to some of these proteins. These include:

Heinz et al. Plasma membrane-located purine nucleotide transport proteins are key components for host exploitation by microsporidian intracellular parasites. *PLoS Pathog.* 2014 Dec 4;10(12):e1004547.

Hacker et al. Strategies for maximizing ATP supply in the microsporidian *Encephalitozoon cuniculi*: direct binding of mitochondria to the parasitophorous vacuole and clustering of the mitochondrial porin VDAC. *Cell Microbiol.* 2014 Apr;16(4):565-79.

Somewhat related to Heinz et al. but with no identified proteins on either the parasite or host side but the data by Hacker et al. suggest specific protein-protein interactions between the host and parasite mediating host mitochondria-parasite interactions. The methodology used here for *C. elegans* would be of great benefit for this mammalian system too.

Zhao et al. Development of a strategy for the identification of surface proteins in the pathogenic microsporidian *Nosema bombycis*. *Parasitology.* 2015 Jun;142(7):865-78.

A complementary proteomics, albeit much simpler, approach was used to investigate the proteins from the spore stage of microsporidia.

Yang et al. Interaction and assembly of two novel proteins in the spore wall of the microsporidian species *Nosema bombycis* and their roles in adherence to and infection of host cells. *Infect Immun*. 2015 Apr;83(4):1715-31.

Polonais et al. Microsporidian polar tube proteins: highly divergent but closely linked genes encode PTP1 and PTP2 in members of the evolutionarily distant *Antonospora* and *Encephalitozoon* groups. *Fungal Genet Biol*. 2005 Sep;42(9):791-803.

3) On page 12, lines 12-14, the authors could cite two relevant, recent, papers relating to the insect *Drosophila* and mammalian cells investigated with the same or related methodology as, indeed, these host model systems will be relevant to study numerous host-microsporidia interactions. More generally this elegant approach will be of great interest to study a broad range of intracellular pathogens including viruses, bacteria and microbial eukaryotes in various hosts.

Chen et al. Proteomic mapping in live *Drosophila* tissues using an engineered ascorbate peroxidase. *Proc Natl Acad Sci U S A*. 2015 Sep 29;112(39):12093-8.

Hung et al. Spatially resolved proteomic mapping in living cells with the engineered peroxidase APEX2. *Nat Protoc*. 2016 Mar;11(3):456-75.

4) Although the authors cite a paper (5) describing a *Toxoplasma* protein targeted to the host nucleus where it alters gene expression, I think that the rationale of investigating potential Nematocida proteins located in the host nucleus should be explicitly stated in the main text. These parasite proteins could contribute at modulating gene expression or other key features of the host nucleus.

Additional points

I could not find any definition for NPN in Fig. S2.

In table S7: Indicate the species for EQB61122.1 and related entries – are these from *Nosema apis* BRL 01?

Cited reference 47 is incomplete, the journal details need to be added.

Reviewer #2 (Remarks to the Author):

Reinke et al used spatially restricted enzymatic tagging and mass spectrometry to examine secreted and surface proteins of Nematocida microsporidia. The authors identified 82 microsporidia inside *C. elegans* intestinal cells, including several pathogen proteins in the nucleus. Further bioinformatics analyses reveal several properties of identified proteins. The ascorbate peroxidase (APEX)-based labeling strategy was originally developed to tag proteins within a confined compartment of interest in order to facilitate subsequent enrichment of those proteins of interest from a complex matrix (e.g., the total cell lysates). Herein the authors applied this approach to analyze secreted and surface proteins of an intracellular pathogen. However, in their case both proteins of interest AND matrix proteins (i.e., host cellular proteins) were tagged, thus essentially no enrichment was achieved in this process. In the manuscript, the authors also predict the presence of a large number of secreted/surface proteins (on the order of 300-900) and yet less than 100 were identified by such a systematic approach, which is largely due to the fact that no enrichment was afforded by their

strategy. Therefore, the described approach barely offers any advantages in terms of analyzing pathogen proteins that interact directly with host cells and the reviewer does not think this work presents any significant technical advances or biological insights. Other minor comments include:

1. Within the identified secreted/surface proteins in this work, are there any proteins known in previous studies? Presence of known secreted/surface proteins in the dataset could add some strength to their findings.
2. Other than bioinformatics analysis, the authors made no experimental attempts to validate the proteomic findings or yield significant new biology regarding host-pathogen interactions.
3. The format of Ref 10 is incorrect. The journal name contains both the full name and abbreviation.

REVIEWERS' COMMENTS:

Reviewer #1 (Remarks to the Author):

The authors have addressed all key issues identified in the earlier version of the manuscript. Hence I have no reservation for the publication of this very interesting and important study.

I have one suggestion for a detail: I would have an additional table listing the entries for which there is evidence for processing of the SP by the SPase, with corresponding annotation of the entries along with the sequences of the peptide(s) supporting the claim of the authors. This would be useful as the additional information relating to this issue that added to the existing tables was not straightforward to understand and the peptides are not shown.

Reviewer #2 (Remarks to the Author):

The authors set out to identify the secreted and surface proteins of microsporidia within infected *C. elegans*. The best approach, in the reviewer's opinion, would be selective labeling of secreted and surface proteins of the pathogen followed by affinity purification. The APEX-based strategy described in the paper allowed targeting to infected intestinal cells, yet it still indiscriminately tagged both pathogen and host proteins. The reality is that within infected host cells the secreted and surface proteins of the pathogen are overwhelmed by exceedingly large amounts of host proteins, making their mass spec detection very challenging. The described approach offers no advantage at all to address this critical issue in identifying pathogen proteins that interact directly with host cells. Thus, the reviewer does not think this work presents a significant technical advance. In addition, there is no biological validation of any findings in the manuscript, leaving most conclusions only descriptive without experimental evidence. The reviewer does not recommend its publication in *Nat Commu*.

We thank the reviewers for their comments. We have provided a revised manuscript (where all changes are highlighted) based on the reviewers concerns. Several points raised by the reviewers have been incorporated into the manuscript and we have provided a point by point response to each of the reviewer's comments:

Reviewer #1 (Remarks to the Author):

In this manuscript the authors adapted for C. elegans an elegant and powerful methodology to differentially tag and identify proteins through proteomics that are located either in the nucleus or the cytosol of the nematode and exploited this approach to identify effector proteins from the microsporidia Nematocida, a natural intracellular pathogen of the nematode. The authors complement their proteomics MS analyses with a number of detailed bioinformatics analyses to further characterise the properties of the proteins of interest and ensure these are indeed strong candidate effector proteins of two Nematocida species. Other new data include the genome sequence of a new species of Nematocida and a global bioinformatics analysis of gene families across available annotated microsporidia genomes. The different claims of the authors are strongly supported by these very well presented data.

This is a very important study, which also represent a technical tour de force, as very little is currently know about effector proteins mediating host-microsporidia interactions and these intracellular pathogens are increasingly recognised as important in animals and humans in a number of ecosystems and medical contexts. More generally this methodology will undoubtedly influence the study of numerous host-intracellular pathogens systems. The paper is very nicely put together presenting experiments carefully performed and data conservatively interpreted. I have only a few details that should be considered to slightly improve the presentation and interpretations of these data and boost the "selling" of the used methodology to study host-intracellular pathogens interactions.

1) A key feature of a subset of the identified proteins is the presence of an N-terminal signal peptide (SP). Is there any evidence from the MS data, for some proteins at least, that the SPs were processed by the RER signal peptidase (SPase), assuming that a SPase is present in Nematocida? This would nicely link the localisation data with the molecular and cellular processes underlying the secretion of these proteins by the pathogen.

The reviewer raises a good point. We have now re-analyzed our mass spectrometry data to identify the signal peptidase-processed forms of proteins with predicted signal peptides. Of the 22 host-exposed proteins we identified with high confidence that have predicted signal peptides, for 4 of these proteins we have now detected peptides that contain the N-terminal fragment that would be produced from cleavage by a signal peptidase. This result provides additional support that these proteins are being secreted into the host. We have added this result in the discussion on page 6, lines 10-17, and Tables S3 and S6.

2) As stated in the manuscript only few microsporidia proteins mediating interactions with their host have been identified. I thought that here the authors could have been more scholarly by citing some additional papers describing these. Currently only two, relevant, papers are cited (23,24). Indeed the presented data, some in the supplementary material, refer to some of these proteins. These include:

Heinz et al. Plasma membrane-located purine nucleotide transport proteins are key components for host exploitation by microsporidian intracellular parasites. *PLoS Pathog.* 2014 Dec 4;10(12):e1004547.

Hacker et al. Strategies for maximizing ATP supply in the microsporidian *Encephalitozoon cuniculi*: direct binding of mitochondria to the parasitophorous vacuole and clustering of the mitochondrial porin VDAC. *Cell Microbiol.* 2014 Apr;16(4):565-79.

Somewhat related to Heinz et al. but with no identified proteins on either the parasite or host side but the data by Hacker et al. suggest specific protein-protein interactions between the host and parasite mediating host mitochondria-parasite interactions. The methodology used here for *C. elegans* would be of great benefit for this mammalian system too.

Zhao et al. Development of a strategy for the identification of surface proteins in the pathogenic microsporidian *Nosema bombycis*. *Parasitology.* 2015 Jun;142(7):865-78.

A complementary proteomics, albeit much simpler, approach was used to investigate the proteins from the spore stage of microsporidia.

Yang et al. Interaction and assembly of two novel proteins in the spore wall of the microsporidian species *Nosema bombycis* and their roles in adherence to and infection of host cells. *Infect Immun.* 2015 Apr;83(4):1715-31.

Polonais et al. Microsporidian polar tube proteins: highly divergent but closely linked genes encode PTP1 and PTP2 in members of the evolutionarily distant *Antonospora* and *Encephalitozoon* groups. *Fungal Genet Biol.* 2005 Sep;42(9):791-803.

We agree with the reviewer that the inclusion of these references strengthen the paper and they have been added to page 3, lines 17-19.

3) On page 12, lines 12-14, the authors could cite two relevant, recent, papers relating to the insect *Drosophila* and mammalian cells investigated with the same or related methodology as, indeed, these host model systems will be relevant to study numerous host-microsporidia interactions. More generally this elegant approach will be of great interest to study a broad range of intracellular pathogens including viruses, bacteria and microbial eukaryotes in various hosts.

Chen et al. Proteomic mapping in live *Drosophila* tissues using an engineered ascorbate peroxidase. *Proc Natl Acad Sci U S A.* 2015 Sep 29;112(39):12093-8.

Hung et al. Spatially resolved proteomic mapping in living cells with the engineered peroxidase APEX2. *Nat Protoc.* 2016 Mar;11(3):456-75.

We have added these references on page 13, line 1-4.

4) Although the authors cite a paper (5) describing a *Toxoplasma* protein targeted to the host nucleus where it alters gene expression, I think that the rationale of investigating potential *Netmatocida* proteins located in the host nucleus should be explicitly stated in the main text.

These parasite proteins could contribute at modulating gene expression or other key features of the host nucleus.

On page 12, lines 20-22 we added a statement about the potential functional of Nematocida proteins localized to the host nucleus.

Additional points

I could not find any definition for NPN in Fig. S2.

We have now explicitly defined NPN in the figure legend for Fig. S2,

In table S7: Indicate the species for EQB61122.1 and related entries – are these from Nosema apis BRL 01?

These entries are from *Nosema apis BRL 01*, and the species name has now been corrected to reflect this.

Cited reference 47 is incomplete, the journal details need to be added.

This reference has now been corrected.

Reviewer #2 (Remarks to the Author):

Reinke et al used spatially restricted enzymatic tagging and mass spectrometry to examine secreted and surface proteins of Nematocida microsporidia. The authors identified 82 microsporidia inside C. elegans intestinal cells, including several pathogen proteins in the nucleus. Further bioinformatics analyses reveal several properties of identified proteins. The ascorbate peroxidase (APEX)-based labeling strategy was originally developed to tag proteins within a confined compartment of interest in order to facilitate subsequent enrichment of those proteins of interest from a complex matrix (e.g., the total cell lysates). Herein the authors applied this approach to analyze secreted and surface proteins of an intracellular pathogen. However, in their case both proteins of interest AND matrix proteins (i.e., host cellular proteins) were tagged, thus essentially no enrichment was achieved in this process. In the manuscript, the authors also predict the presence of a large number of secreted/surface proteins (on the order of 300-900) and yet less than 100 were identified by such a systematic approach, which is largely due to the fact that no enrichment was afforded by their strategy. Therefore, the described approach barely offers any advantages in terms of analyzing pathogen proteins that interact directly with host cells and the reviewer does not think this work presents any significant technical advances or biological insights. Other minor comments include:

The reviewer correctly points out that the advantage of the APX-based labeling strategy is the ability to enrich for proteins in a specific location. This type of specific labeling is what we set out to achieve by targeting APX to specific tissues to label proteins present in either the intestinal cytoplasm or the intestinal nucleus of *C. elegans* that were infected with microsporidia. We hypothesized that these locations would contain not only host cellular proteins, but also microsporidia proteins. This is indeed what we found by identifying 891 *C. elegans* proteins and 82 microsporidia proteins that are statistically enriched over control experiments consisting of animals that lacked the APX enzyme. We identified a number of *C. elegans* proteins specific to

either the nucleus or cytoplasm, and these sets of proteins are significantly enriched in proteins known to be localized to those subcellular compartments (Figure S4). The microsporidia proteins are significantly enriched in a number of properties including targeting signals that we hypothesize allow these proteins to be localized to be in direct contact with the host (Figure 2B).

Expecting there to be enrichment of microsporidia proteins over host cellular proteins is an incorrect interpretation of what is occurring biologically. Both host cellular proteins and microsporidia host-exposed proteins will be present in the cytoplasm or the nucleus. In fact, the main advantage of our approach is that it allows for the identification of these microsporidia proteins even when the majority of the proteins present in the location are from the host. Additionally, the presence of host cellular proteins provides powerful internal controls (Figure S4).

Based on the fraction of the *C. elegans* intestinal proteome identified, which we estimate to be 8-24%, we propose that there are 300-900 host-exposed *N. parisii* proteins. Because we are identifying less than a quarter of the intestinal proteome, we would not expect to identify all of the microsporidia host-exposed proteins. Although the 82 proteins we identified are only a fraction of all Nematocida host-exposed proteins, we believe this study represents the largest systematic identification of host-exposed proteins in an animal to date. Thus, this study provides a significant technical advance in the ability to identify these putative pathogen effector proteins. We identified a number of properties that characterize host-exposed proteins including the observation that they are composed of many paralogous proteins and are rapidly evolving. These results reveal important biological insight into how these pathogens are likely evolving to interact with their hosts.

1. Within the identified secreted/surface proteins in this work, are there any proteins known in previous studies? Presence of known secreted/surface proteins in the dataset could add some strength to their findings.

There have been no previous identifications of any *Nematocida* secreted or surface proteins to compare our data set with. Several of these types of proteins have been identified in other microsporidia species, but the proteins we identified are significantly enriched in proteins not conserved with other microsporidia species. We did identify one protein, hexokinase, in both our *N. parisii* and *N. sp 1* experiments. Microsporidia hexokinases were previously shown to be secreted in a heterologous system (Cuomo et al) and were experimentally confirmed to be host-exposed in *Antonospora locustae* infections (Senderskiy et al). These points are detailed in the discussion on page 14 lines 11-16.

2. Other than bioinformatics analysis, the authors made no experimental attempts to validate the proteomic findings or yield significant new biology regarding host-pathogen interactions.

Although we were not able to validate the location of our identified host-exposed proteins, our experiments were able to identify *C. elegans* proteins previously shown to be localized to the nucleus and cytoplasm, supporting the specificity of the technique. Additionally, the identified microsporidia host-exposed proteins are significantly enriched in a number of properties including transmembrane domains and signal peptides. Although mechanistic studies on the biological functions of these proteins is beyond the scope of this study, our identification and prediction of host-exposed proteins provides a powerful resource for further work determining the biochemical and biological mechanisms of these proteins.

3. *The format of Ref 10 is incorrect. The journal name contains both the full name and abbreviation.*

This reference has now been corrected.

We thank the reviewers for their comments. We have provided a revised manuscript (where all changes have been tracked) based on the reviewers concerns. Below we provide a point by point response to each of the reviewer's comments:

Reviewer #1 (Remarks to the Author):

The authors have addressed all key issues identified in the earlier version of the manuscript. Hence I have no reservation for the publication of this very interesting and important study.

I have one suggestion for a detail: I would have an additional table listing the entries for which there is evidence for processing of the SP by the SPase, with corresponding annotation of the entries along with the sequences of the peptide(s) supporting the claim of the authors. This would be useful as the additional information relating to this issue that added to the existing tables was not straightforward to understand and the peptides are not shown.

We have added an additional table (Supplementary Table 2) that includes the sequence and peptide counts of the N-terminal peptides identified.

Reviewer #2 (Remarks to the Author):

*The authors set out to identify the secreted and surface proteins of microsporidia within infected *C. elegans*. The best approach, in the reviewer's opinion, would be selective labeling of secreted and surface proteins of the pathogen followed by affinity purification. The APEX-based strategy described in the paper allowed targeting to infected intestinal cells, yet it still indiscriminately tagged both pathogen and host proteins. The reality is that within infected host cells the secreted and surface proteins of the pathogen are overwhelmed by exceedingly large amounts of host proteins, making their mass spec detection very challenging. The described approach offers no advantage at all to address this critical issue in identifying pathogen proteins that interact directly with host cells. Thus, the reviewer does not think this work presents a significant technical advance. In addition, there is no biological validation of any findings in the manuscript, leaving most conclusions only descriptive without experimental evidence. The reviewer does not recommend its publication in Nat Commu.*

We respectively disagree with this reviewer's comments as we believe our approach of tagging pathogen proteins from infected cells in an intact animal is a clear advance over existing approaches. Although it is potentially much simpler to identify the pathogen proteins outside of the host, the microsporidian pathogens we studied are not possible to grow outside of a host. This is true for many pathogens where our methodology is applicable. Additionally, even for pathogens that can be grown outside of a host, the proteins that are on the surface or are secreted might be different than when the pathogen is growing in a host context. Indeed, we did identify many host proteins in our experiments, and these proteins provided an important internal control to demonstrate that our method identified proteins with location specificity. Identifying these host proteins did not prevent us from identifying pathogen proteins in specific locations. Importantly, we identified 82 pathogen proteins with high confidence that were statistically enriched for a number of properties, including signal peptides and rapid evolution.